# Trends in the Prescription of Strong Opioids for Chronic Non-Cancer Pain in Primary Care in Catalonia: Opicat-Padris-Project

**DOI:** 10.3390/pharmaceutics14020237

**Published:** 2022-01-20

**Authors:** Aina Perelló-Bratescu, Christian Dürsteler, Maria Asunción Álvarez-Carrera, Laura Granés, Belchin Kostov, Antoni Sisó-Almirall

**Affiliations:** 1Larrard Primary Health Center, Parc Sanitari Pere Virgili, 08024 Barcelona, Spain; 2Pain Medicine Section, Anaesthesiology Department, Hospital Clínic de Barcelona, 08036 Barcelona, Spain; dursteler@clinic.cat; 3Pain Medicine Section, Surgery Department, Hospital Clínic de Barcelona, 08036 Barcelona, Spain; 4Pharmacy Service, Parc Sanitari Pere Virgili, 08024 Barcelona, Spain; aalvarez@perevirgili.cat; 5Preventive Medicine and Epidemiology Department, Hospital Clínic de Barcelona, 08036 Barcelona, Spain; granes@clinic.cat; 6Primary Healthcare Transversal Research Group, Institut D’Investigacions Biomèdiques August Pi i Sunyer (IDIBAPS), 08036 Barcelona, Spain; badriyan@clinic.cat; 7Les Corts Primary Health Care Center, Consorci d’Atenció Primària de Salut Barcelona Esquerra (CAPSBE), 08029 Barcelona, Spain; asiso@clinic.cat

**Keywords:** analgesics, opioids, opioid-related disorders, inappropriate prescribing, chronic pain, physicians, primary care, big data, pharmacoepidemiology, pharmacoeconomics

## Abstract

In chronic non-cancer pain (CNCP), evidence of the effectiveness of strong opioids (SO) is very limited. Despite this, their use is increasingly common. To examine SO prescriptions, we designed a descriptive, longitudinal, retrospective population-based study, including patients aged ≥15 years prescribed SO for ≥3 months continuously in 2013–2017 for CNCP in primary care in Catalonia. Of the 22,691 patients included, 17,509 (77.2%) were women, 10,585 (46.6%) were aged >80 years, and most had incomes of <€18,000 per year. The most common diagnoses were musculoskeletal diseases and psychiatric disorders. There was a predominance of transdermal fentanyl in the defined daily dose (DDD) per thousand inhabitants/day, with the greatest increase for tapentadol (312% increase). There was an increase of 66.89% in total DDD per thousand inhabitants/day for SO between 2013 (0.737) and 2017 (1.230). The mean daily oral morphine equivalent dose/day dispensed for all drugs was 83.09 mg. Transdermal fentanyl and immediate transmucosal release were the largest cost components. In conclusion, there was a sustained increase in the prescription of SO for CNCP, at high doses, and in mainly elderly patients, predominantly low-income women. The new SO are displacing other drugs.

## 1. Introduction

The World Health Organization (WHO) recognizes pain as a major public health problem. The main causes of chronic non-cancer pain (CNCP) are musculoskeletal diseases, one of the primary causes of chronic disease (first in women and second in men). Spanish surveys conducted in 2017 and 2020 [1,2] found that 25% of the population had chronic pain, and the prevalence was higher the lower the social class. Pain is the second cause of primary care (PC) consultation and >50% of appointments are related to chronic pain [3]. Widespread chronic pain worsens the quality of life, mental health and the cardiovascular risk, and increases health spending [4]. CNCP is a multidimensional problem, requiring a multifactorial approach. More pain drugs are consumed than any other class of drug [5], including strong opioids (SO) (morphine, fentanyl, hydromorphone, oxycodone, tapentadol and buprenorphine) of proven efficacy and with indications in acute and cancer pain [6]. The use of SO increases with age [7] and older people are especially sensitive to the adverse effects, given the comorbidity and use of concomitant medication they usually present [8]. In CNCP, evidence of the effectiveness of SO is very limited [9,10], so current clinical guidelines discourage their widespread use for CNCP and recommend evaluating non-pharmacological alternatives prior to the use of SO [11,12].

The responsible prescription of SO for CNCP requires evaluation of the balance of potential benefits and risks, as well as close monitoring of their possible adverse effects and the clinical response [13]. Some of these risks include addiction and abuse, tolerance and overdose [13,14,15,16], depression [17], respiratory depression and death [18].

However, SO-related deaths are not diagnosed as such in many cases, underestimating the real dimension of the risk. In many developed countries, opioid-related deaths are a real epidemic; in the US there was a 90% increase in SO-related deaths between 2013 and 2017 [19].

Despite the recommendations, the use of SO for CNCP is increasingly common in our setting [20]. Data on the dispensing of SO in Spain show a sustained increase in their use in recent years (44% increase in 2013–2017). 

The objective of this study was to describe PC prescriptions of SO for CNCP over a 5 year period detecting practices considered risky and analysing their causes.

## 2. Materials and Methods

### 2.1. Design

We made a descriptive, longitudinal retrospective study with data obtained from the Health Research and Innovation Data Analysis Programme (PADRIS), of the Health Department of the Government of Catalonia (the Catalan Health Surveillance System, CHSS), whose aim is to make health data available to the scientific community to promote health research, innovation, and evaluation through access to the reuse of health data generated by Catalonia’s comprehensive public health care service. 

PADRIS periodically collects detailed individual-level information on demographic and socioeconomic characteristics, and exhaustive health-related and medical resource use information generated by the interactions between users and the public healthcare system that provides universal public health coverage to all Catalan residents. This longitudinal, quality-controlled, updated information system permits epidemiological analyses, evaluations of healthcare interventions and programs, and public analysis and benchmarking of health indicators across healthcare areas, among other assessments. Specifically, for healthcare-related data, the database has, since 2012, progressively collected information from several sources, including the Minimum Basic Dataset for Healthcare Units registry (which includes hospitalizations, primary care visits, and skilled nursing facility visits), information on pharmacy prescription fillings, and billing records, which include hospital outpatient clinic visits to specialists and emergency department visits. The Register includes an automated validation system, and regular external audits are carried out. More information is available at: https://aquas.gencat.cat/ca/ambits/analitica-dades/padris/ (accessed on 10 January 2022).

The data obtained were anonymised, eliminating from the database any potential da-ta that could identify patients. The data request was made in December 2018. Previously, an agreement was signed for the transfer of anonymised health data between the Agency for Health Evaluation and Quality of Catalonia (AQuAS), which is responsible for the PADRIS programme, and the investigators. The first database was provided by AQuAS in March 2019 and the final database, following validation, was received in February 2020.

### 2.2. Study Population

We included all inhabitants inscribed in the Catalonian public health system, which had an assigned population of 7,753,482 in 2013, and 7,628,166 in 2017. All inhabitants aged ≥15 years prescribed a SO by PC physicians for ≥ 3 months continuously in the previous five years (2013–2017) were selected. The duration of the SO prescription for ≥3 months was based on the definition of chronic pain in the current guidelines as pain that persists or recurs for more than 3 months [3,21]. To ensure SO were not prescribed for oncological pain, all patients diagnosed with cancer (according to ICD-9 and ICD-10 encoding) in the previous 5 years, and patients receiving parenteral opioids (except transdermal opioids) were excluded.

### 2.3. Variables Included

Demographic data (age, sex, socioeconomic level, and health care region [in 2017]).

DDDs (defined daily dose) per 1000 inhabitants-days. The PADRIS database includes information on the number of DDD (https://www.who.int/tools/atc-ddd-toolkit/about-ddd, accessed on 10 January 2022) dispensed monthly per inhabitant of each SO, according to the Anatomical Therapeutic Chemical (ATC) classification coding. In the case of fentanyl and buprenorphine patches, the DDD provided corresponds to mcg/hour, and was thus multiplied by 72 h (standard duration of the patches). To calculate DDDs per 1000 inhabitant-days, the number of annual DDD for each ATC code was summed. This was divided by the total number of inhabitants inscribed in the Catalan Health System according to PADRIS, and by 365 days, and was multiplied by 1000.

Daily oral morphine milligram equivalent dose (MME): from the number of DDD dispensed monthly, the number of mg/day/inhabitant of each SO was calculated, assuming that each dispensation corresponds to one month of treatment as established in the electronic prescription system. Milligrams were transformed into MME by applying the conversion factor of the Oral Opioid Morphine Milligram Equivalent (MME) conversion factors table [22].

Cost associated with the SO prescription.

Active diagnoses during the study period of diseases potentially responsible for CNCP (musculoskeletal disease, herpes zoster, trigeminal neuralgia, fibromyalgia) and mental disorders.

### 2.4. Ethical Aspects

The procedures followed Spanish and Catalan laws. Researchers followed the ethical standards of the Declaration of Helsinki for biomedical studies and the activities de-scribed followed the Code of Good Practice in clinical research. 

The data were anonymised, and the study protocol was presented and approved by the Research Ethics Committee of the Hospital Clinic of Barcelona on 26 September 2018 (Ref. HCB/2018/0749).

### 2.5. Statistical Analysis

Categorical variables were presented as absolute frequencies and percentages, and continuous variables as the mean and standard deviation (SD). The statistical analysis was made using R version 3.6.1 for Windows (R Foundation for Statistical Computing, Vienna, Austria, 2021).

## 3. Results

We included 22,691 patients, of whom 17,509 (77.2%) were women. The patients included correspond to 0.29% of the total inscribed population in 2017. 

### 3.1. Demographic Distribution

The distribution by age, health region and socio-economic level is shown in Table 1: 15,389 (67.7%) patients were aged >70 years and 10,585 (46.6%) >80 years.

The most common diagnoses were musculoskeletal diseases (in all patients), psychiatric disorders in 11,475 (50.6%), diseases causing neuropathic pain in 3866 (17%) and fibromyalgia in 915 (4%).

### 3.2. DDD

Analysis of the evolution of DDD per 1000 inhabitant-days of the SO dispensed (Figure 1) showed a predominance of the prescription of transdermal fentanyl, with an increase of 63.5% in the DDD per 1000 inhabitant-days between 2013 (0.453) and 2017 (0.741). The largest increase was for tapentadol, with an increase of 312% in the DDD per 1000 inhabitant-days (0.032 in 2013 to 0.132 in 2017). There was an increase in the prescription of virtually all SO analysed (except hydromorphone), with a total increase of 66.89% in total DDD per 1000 inhabitant-days between 2013 (0.737) and 2017 (1.230).

### 3.3. MME

The mean MME/day dispensed for all SO was 83.09 mg. There were 383,842 (59.86%) dispensations with a MME/day of >50 mg, and 198,477 (30.95%) with a MME/day of >90 mg. A total of 19,804 (87.4%) patients received a MME/day of >50 mg and 14,365 (63.4%) of >90 mg. Analysis of MME/day >50 mg by SO (Table 2) showed most (250,456, 65.25%) were for transdermal fentanyl, as were those for a MME/day >90 mg (139,081 transdermal fentanyl, 70%). The mean dose was 106 mg MME/day for fentanyl, 84 mg for buprenorphine and 70 mg for tapentadol.

### 3.4. Economic Impact

Transdermal fentanyl and transmucosal immediate release fentanyl (TIRF) were the main cost components. Expenditure on TIRF rose from €2,654,919 in 2013 to €4,097,149 in 2017, an increase of 54%, and that on transdermal fentanyl rose from €2,522,131 in 2013, to €4,033,144 in 2017, an increase of 59.9%. Expenditure on tapentadol increased by 303% (from €518,211 in 2013, to €2,093,121 in 2017). Total spending on SO increased by 60.88% in the study period. Analysis of the total amount spent on SO during the study period (€54,380,141.76), shows that transdermal fentanyl accounted for 31.87% (€17,334,211) of the total, TIRF for 31.76% (€17,274,684), oxycodone for 13.9% (€7,569,059) and tapentadol for 12.75% (€6,934,209) (Figure 2).

## 4. Discussion

This is the first study to analyse the prescription of SO for CNCP by PC physicians over a 5 year period through the PADRIS program using data mining and big data. The vast majority of the 22,691 patients were women from urban areas with a low socioeconomic status and advanced age, and with musculoskeletal diseases. There was a large increase in SO DDDs per 1000 inhabitant-days in the study period, with a predominance of transdermal fentanyl and a significant increase in tapentadol prescriptions. Spending on SO increased considerably during the study period, with transdermal fentanyl and TIRF accounting for most of the increase. Most patients received a MME/day of >50 mg, and more than half received a MME/day of >90 mg. 

Of the 22,691 patients, 77.2% were women, in agreement with some studies [23], although others found no significant difference between the sexes (49–59% of men in the study by Khalid [24]. A female predominance may be explained by the results of population surveys on perceived health [1,2].

Nearly half the prescriptions were for people aged >80 years. This is consistent with the increase in pain at these ages according to the surveys cited above, and the tendency to increase the prescription of SO according to age is consistent with previous studies [7]. This increase is worrying when the risks of prescribing SO at these ages are considered. There is little or no evidence on the effects of SO in older people, as they are virtually unrepresented in clinical trials, and the likelihood of adverse effects increases (especially falls, sedation, overdose, and an increased risk of respiratory depression or cognitive impairment), in tandem with comorbidity and polypharmacy [8,25].

Most patients had an income of <€18,000 per year. This is consistent with increased pain in low-income populations [1,2] and with previous studies in which the relationship between SO consumption and low income is proven, such as that by Friedman [26]. Although the association between low incomes and pain is established in Catalonia [1], the factors that predispose people with low (or very low) incomes to be treated with SO in the context of CNCP have not yet been well studied. 

With respect to the geographical distribution, one third of requirements for SO occurred in semi-urban and rural areas. Studies such as that by Prunuske [23] found an increase in the prescription of SO associated with rurality. Taking into account the fact that the urban population accounts for about half the population of Catalonia, more studies are needed to determine how rurality influences the prescription of SO.

Half the patients had been diagnosed with psychiatric disorders, anxiety or depression, coinciding with previous studies, such as those by Reid [27] and Khalid [24]. Psychiatric disorders are commonly observed in patients with CNCP, which is often a cause. However, SO, especially at high doses [28] promote the onset of depressive disorders. This is a vicious circle that is difficult to break, but has important associated risks, such as the concomitant prescription of anxiolytics and SO.

Four percent of patients had a diagnosis of fibromyalgia, in which the use of SO is discouraged [11] and 17% had neuropathic pain, in which SO should be used as third-line treatments [5,10,13,29]. Although a table with indications for SO prescriptions in CNCP would have been useful, this would be difficult to draw up. Strong opioids are not recommended by almost all clinical practice guidelines for the treatment of CNCP [9,10,11,12]. The only case of CNCP for which treatment with SO is recommended is neuropathic pain, with a weak recommendation according to the GRADE Classification [29].

Analysis of DDDs per 1000 inhabitant-days and their evolution shows there was a greater increase in SO DDDs per 1000 inhabitant-days in our study population than that found by other studies [20], probably because our patients were older and more complex clinically. This is particularly worrying, as the use of SO in CNCP lacks evidence [10].

The predominance of fentanyl prescription over the other SO is not justified by guideline recommendations, which state that the superiority of fentanyl over the other SO has not been demonstrated [6,7] and shows a differing pattern to other European countries and the USA [18]. In fact, transdermal fentanyl is not recommended as a starting drug in opioid-naïve patients [30,31], which was very common in our setting [32]. The specific pharmaceutics of transdermal fentanyl, which facilitates treatment adherence, could partly explain the predilection of physicians and patients for it [33,34,35,36,37,38]. Other factors, such as the commercial pressure exerted on physicians in the 2000s, and the progressive normalization of SO use for chronic pain treatment by physicians and the public recently, could also have influenced the exponential increase in the prescription of transdermal fentanyl.

The increase in tapentadol prescriptions is surprising. There are few clinical trials comparing the efficacy of tapentadol with other SO and most are of low quality, as are studies on the long-term safety and efficacy of tapentadol [5,7,39]. In fact, France (2017) and Canada (2018) decided to stop funding tapentadol for CNCP because of the lack of evidence. We still do not know the relevance of the increasing prescription of new opioids in clinical practice, as there are no studies published. 

Analysis of the MME dispensed, despite its limitations [40] showed that most patients received doses of >50 mg MME/day, and more than half >90 mg. These doses are more typical of the treatment of cancer pain and palliative care [41,42]. Their use in elderly patients with serious comorbidities, is contra-indicated from the point of view of patient safety [8,25,43]. Scherrer [28] reported that initiating treatment at 50 mg MME/day increases the likelihood of adverse effects such as depression, and requires intensified follow-up, while there is a consensus that doses of 90 mg MME/day [5] should be avoided due to the risk of death by overdose. The mean dose of transdermal fentanyl dispensed is >90 mg MME/day.

The medicalization of CNCP, with the consequent increase in SO prescriptions, may be due to various reasons: difficult access to non-pharmacological pain treatment, which could be improved by introducing more physiotherapy and psychoeducation in primary care, the demand for immediate solutions to pain and the high expectations of drug effectiveness and their use to relieve emotional discomfort. There are several studies where the impact of non-pharmacological treatment is proven in CNCP [44,45,46,47], therefore it should be a first line treatment.

The shortfall in the training of professionals involved in CNCP management, as shown in previous studies [48], may also play a role in the serious deviations observed in the prescription of SO in Catalonia.

Likewise, the increase in some SO, such as tapentadol and oxycodone, without scientific justification, is a response to the powerful advertising and marketing strategies which, as happened in the USA with oxycodone, show their influence and the associated risks [49].

### Limitations of the Study

Analysis of associated diagnoses in patients prescribed SO is limited. It is not possible to determine the exact disease for which SO are prescribed (at the time of writing it was not mandatory to associate the prescription with a diagnosis). This limitation affects the data extraction as well, as we were not able to associate SO prescription with a CNCP diagnosis. We decided that the best solution to address this issue was to underestimate our findings, removing patients with a cancer diagnosis even if they had SO prescribed for CNCP, instead of overestimating them by adding cancer patients treated with SO for their neoplasm, and not for a CNCP condition.

## 5. Conclusions

Strong opioid use increased steadily during the study period, mostly in vulnerable patients, (elderly women with low incomes) at high or very high doses. The new SO are displacing other drugs.

## Figures and Tables

**Figure 1 pharmaceutics-14-00237-f001:**
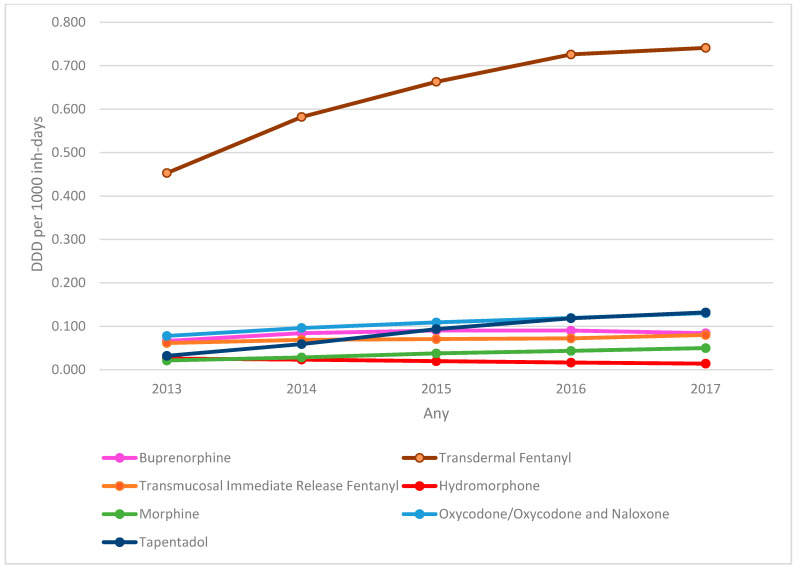
Evolution of strong opioids DDD per 1000 inhabitant-days by active substance. DDD: defined daily dose.

**Figure 2 pharmaceutics-14-00237-f002:**
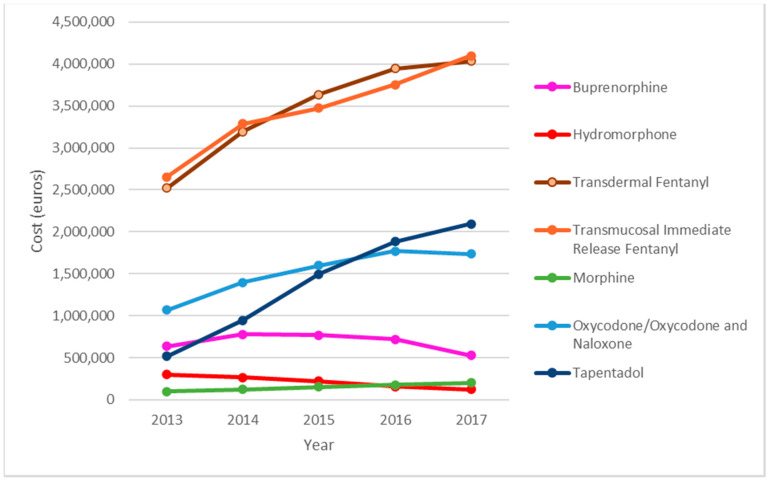
Evolution of the economic cost by type of strong opioids.

**Table 1 pharmaceutics-14-00237-t001:** Baseline characteristics of the study population.

Variable	No. 22691
Sex	
Female	17,509 (77.2%)
Male	5182 (22.8%)
Age	
<30 years	29 (0.1%)
30–39 years	353 (1.6%)
40–49 years	1430 (6.3%)
50–59 years	2505 (11.0%)
60–69 years	2985 (13.2%)
70–79 years	4804 (21.2%)
80–89 years	7426 (32.7%)
>90 years	3159 (13.9%)
Geographic distribution	
Urban	14,452 (63.7%)
Semiurban	5557 (24.5%)
Rural	2679 (11.8%)
Socioeconomic level	
Exempt from taxation	1368 (6.0%)
<€18,000	18,388 (81.1%)
€18,001–€100,000	2908 (12.8%)
>€100,000	27 (0.1%)

**Table 2 pharmaceutics-14-00237-t002:** Dispensations of >50 mg and >90 mg MME/day by drug.

Drug	MME/Day (mg)	Dispensations			
		>50 mg MME/Day		>90 mg MME/Day	
		(N-383842)		(N-198477)	
	Average ± SD	N	%	N	%
Buprenorphine	84.03 ± 51.15	31,173	8.12	17,660	8.9
Transdermal Fentanyl	106.90 ± 94.41	250,456	65.25	139,081	70.07
Hydromorphone	61.99 ± 63.31	5386	1.4	2186	1.1
Morphine	48.76 ± 80.34	8740	2.28	4447	2.24
Oxycodone/Oxycodone and Nalox-one	51.02 ± 53.30	47,043	12.26	17,417	8.78
Tapentadol	70.74 ± 58.13	41,044	10.69	17,686	8.91

MME: morphine milligram equivalent.

## Data Availability

The data from the Padris program are not available for public.

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
