# Peer review of "Trends in the Prescription of Strong Opioids for Chronic Non-Cancer Pain in Primary Care in Catalonia: Opicat-Padris-Project"

_pharmaceutics, 2022, doi:10.3390/pharmaceutics14020237_

Round 1

Reviewer 1 Report

This study on the use of SO in treatment of CNCP provided very interesting results, which are at the same time very worrisome. It also opened a lot of questions for future research. My suggestions for manuscript improvement are:

  1. Below each table/figure should be a legend
  2. Figure 1. - line colors for TIRF and TDF are very similar, I would suggest a different color for one of them
  3. Consider adding a table with indications for SO prescriptions

Reviewer 2 Report

The author aim to determine the prescription characteristics of CNCP?

The authors reported that:

  • SO prescriptions increase overall with mean MME/day of 83.09 mg. The most commonly prescribing SO is fentanyl patch.
  • 46% of patients are >80 yr; Most common diagnoses are musculoskeletal diseases and 50% are diagnosed with psychological disorders.
  • 3% of patients received an MME/day of > 50 mg, and 65.25% of the prescriptions are fentanyl patch.
  • For prescriptions of MME/day > 90 mg, 70% of the high dose prescriptions are fentanyl patch with a mean dose is 105 mg MME/day (line 160 but in table 2, it is 106.9).
  • Tapentadol is the largest increase in SO prescription within the 5-year timeframe (2013-2017) with mean MME/day of 70.74 mg.
  • SO costs rise 60.88% (TIRF 54%, fentanyl patch 59.9%, tapentadol 303%) within 5 years.
  • 63% of the total cost derives from TIRF and fentanyl patch.
  • Expected DDD per 1000 inhabitant-days in 2030 is 1.7 DDD for fentanyl patch and 0.475 for tapentadol if the trend remains the same as that of 2013-2017

These are all descriptive, which only contribute little to current understanding. Linear regression analysis for the expected DDD per 1000 inhabitant-days in 2030 is simplistic and oversimplify. Actual modeling may help strengthen the work.  

Reviewer 3 Report

Line 67 70 

This paragrqph is a kind mini abstract please focus only on primary and secondary objective do not display the findings of the study 

 Line  90 Please cite the exact definition of chronic pain which will be a reminder for the average reader 

Line 92-94  Please explain better  how non cancer pain were dissociated in patients having cancer 

The tables and figures need more explanatory legends in the bothtom  ex please detail ccp   , accm etc..

This should available all along the paper  abbreviation should be explained clearly before appearing 

Please detail the extraction method of data , who performed it , how did you deal with missing  data , the percentage of missing data , was there any redundance ? may be it is detailed in the PADRIS project. 

Why some data are displayed in 5 years and others displayed for 10 years , please be coherent

Please attribute a mini chapter  for the limitations of ths study 

Line 235 236 please provide reference  PPR

Line 248 250 PPR

Round 2

Reviewer 2 Report

No further comments.

Author Response

Thanks for the review, changes recommended in round 1 have been added

Reviewer 3 Report

Thye authors have responded adequately to all my queries 

Author Response

Thank you for the suggestions, changes have been added in round 1 according to your recommendations